# Navigating the paradigm shift of sex inclusive preclinical research and lessons learnt

Natasha A. Karp

After thirty years of research highlighting the risk of the sex bias in preclinical research, we now have tangible change happening in the research landscape with a rapid increase in the proportions of studies including female and male subjects. In practical terms, this will require a paradigm shift affecting choices around experimental design, analysis and presentation and considerations when drawing conclusions and planning next steps in the pipeline. As the preclinical research community embraces this new way of working, numerous insights are being published and shared on how to proceed. This article will signpost the learnings obtained and potential future steps that should be focused on.

There is a culturally embedded practice of studying only one sex in biology and health research and then generalising to the other, unstudied population[1]. This strategy introduces a knowledge bias into the research pipeline and contributes to translation and reproducibility issues[2–4]. Despite numerous funding bodies establishing initiatives to encourage researchers to routinely integrate males and females into basic, preclinical, and clinical research, little progress has been observed[5]. Consequently, numerous funding bodies have recently issued inclusion mandates requiring justification for exclusion in preclinical research[5–7]. These mandates have increased the proportion of published papers including females and males[8–10] however inclusion still represents a minority of published studies[9]. Additionally, studies including male and female samples are frequently beset by further critical methodological problems that compromise the integrity of their conclusions[8,11]. There is some debate in preclinical research around the choice of prioritising sex as a source of variation over other potential sources of biological variation (e.g., age or genetic background)[12]. In terms of improving reproducibility, embracing biological variation in the designs has been highlighted as the critical step to improve the generalisability of research conclusions[13]. However, research has not found one critical variable that needs to be embraced to achieve this[14]. Some have argued that researchers should be allowed to choose the most relevant source of variation for their research question[12,15]. I would argue that multiple sources of variation can be encompassed in a research study and including female and male samples in preclinical research is an easy first-step to improving the

generalisability. Furthermore, it is an ethical choice as it does not need an increase in sample size by default[16] and uses all animals generated in the breeding process[1].

The strategy of studying only one sex was not a decision taken due to researchers perceiving sex was not relevant or that a particular sex was more valuable than another; rather, it was an ethical and pragmatic decision based on the understanding at that time[17]. Research has shown that scientists believe sex matters but are not running inclusive designs because it was not perceived as achievable[16–18] due to escalating costs and ethical concerns. Many of these perceived barriers are now understood to be misconceptions[16,17] and to change the status quo, there is a need to focus on cultural change[17,19]. For inclusive designs to be the norm, the research community needs a paradigm shift in how research is conducted. This isn't a minor change in how research is conducted, as it requires the community to embrace changes in the language used, the design of experiments, the conduct of experiments, the analysis and presentation of data and the integration of data into in silico models. This article will take the opportunity to showcase some of the steps various researchers have taken to support the transition, and the lessons learnt. It will highlight some of the gaps that are apparent, though, of course, this is based on what is currently understood. It is an exciting time as we traverse this paradigm shift.

## Lessons learnt and opportunities

**Language use within sex inclusive preclinical research.** Culture can be defined as the ideas, customs, and social behaviours of a particular group or society. As we conduct science and use language to communicate, the language used expresses and embodies cultural reality. This matters as the language chosen influences how people conceive of concepts and objects[20].

Frequently, in the context of in vivo research, the word gender has been used as an alternative to sex due to people being uncomfortable with the word sex due to its cultural connotations when used as a shortened form for sexual intercourse. However, the words are not synonyms. In science, sex refers to a classification system that utilises a set of physiological attributes (e.g., chromosomes, hormones, reproductive organs) to define an organism as male, female, intersexed or hermaphrodite. Whilst gender refers to a human-specific phenomenon, arising from the socially constructed roles, expressions, behaviours and identities of female, male and gender-diverse people[21]. In fact, with in vivo, ex vivo or in vitro research it is only possible to explore sex-related variation as the experimental construct cannot be used as a model to explore the human experience of gender. As these terms have very different meanings, when presenting research, it is essential to accurately represent the research and use the correct terminology to avoid confusion[22].

Focusing further on the term sex, it is often presented as a binary truth with phrases such as 'the two sexes' or 'both sexes'. This aligns with a societal culture which is structured around the concept that sex is a binary, biological truth which are mutually exclusive[23]. In reality, for both sex and gender,

these are complex, variable and non-binary classification systems[23]. For example, it is estimated that roughly 5–6% of animals' species are hermaphroditic[24]. Velocci[23] asks whether sex is a useful category and highlights that sex is an incoherent category with no universally agreed guidelines for the definition of sex. Historically, reporting guidelines have focused on reporting the sex and number of animals used in published research[25]. Attention is now focusing on the need to operationalize sex by defining and reporting the variables that were used to distinguish males and females[21,26]. For example, an assessment of chromosomes or hormone concentrations or a visual assessment of primary and secondary sexual organs. Doing so will increase reproducibility[21] and enable the opportunity to consider sex as a constructed category which could operate at multiple biological levels which would then have the potential to "open up new avenues for enquiry in our study of biological variation"[26].

As research practices become more inclusive and identify situations where sex mathematically explains variation in the treatment effect, it is important to be mindful of the trap of perceiving sex as an underlying causal mechanism[21]. When conducting statistical analysis, the term effect is used for the various elements of the model, which normally implies causality but in the context of statistically modelling means that the variation in the data associated with this term is explored. Sex is a category that is represented by multiple mechanisms (hormonal, immune, body composition differences etc.) and therefore it is not sex itself that drives sex-related variation but rather one or more mechanisms associated with sex. Frequently, the language used is reinforcing cultural beliefs that females and males are 'profoundly and systematically different'[27]. For example, I myself used the term 'sexual dimorphism'[2] in a study when describing the prevalence of a treatment effect interacting with sex. In the paper, whilst sex explained variation in the treatment effect, the distributions of the continuous variables still overlapped and yet a term that officially means distinct phenotypic forms was used[2]. Pape et al., also highlighted the frequent inappropriate use of the phrase 'sex-specific' which implies an effect occurs exclusively in one sex category and not the other when more commonly a statistical difference in the degree of the effect is observed[21].

We also need to think about language from a position of inclusion and representation. Research is conducted to aid all and therefore using language such as 'two sexes' or 'both sexes' unintentionally excluding a significant proportion of society. One could argue this is not just an issue that could be accused of wokeism; it is important for challenging cultural biases that can impact the perception of the science being conducted and the conclusions drawn. For example, in 1910, during an expedition to explore Antarctica, George Murray Levick, a zoologist recorded many same-sex mating events whilst studying a penguin colony however, none of these notes would appear in Levick's published work[28]. To address this, it is necessary to be more explicit by stating the sexes used and to avoid an implicit hierarchy by altering the order each time the sexes are declared.

Inclusive research is new territory for many. It is likely that the imprecise use of language to date has arisen from the ongoing paradigm shift. As the community raises these issues, it is important to embrace the learnings and work out how to address them.

### Experimental design

**When do we need a sex inclusive design?.** The Sex and Gender Equity in Research (SAGER) guidelines[21] were published in 2016 to guide the reporting of sex and gender information in study design, data analyses, results and interpretation of findings. They shared a general principle that provides clarity on when a sex inclusive design should be conducted: "Where the subjects of research comprise organisms capable of differentiation by sex, the research should be designed and conducted in a way that can reveal sex-related differences in the results, even if these were not initially expected.". It is a default position of inclusion where the sex can be determined. This means ex vivo samples, primary cell lines and patient-derived cell lines are in scope. Whilst research studies collecting data at the level of a cluster of animals (e.g., a herd or litter group) are exempt. Consensus has not yet been reached on whether immortalised cell lines are in scope, probably as it will depend on the situation. Many researchers have raised challenges with identifying the sex of immortalised cell lines due to contamination with micro-organisms and the loss of Y chromosome due to genetic instability[29]. Holland and Bradbury argued that whilst it is difficult, it is essential to understand and report your findings even if that means being explicit that you could not determine the sex[30]. With increasing update/adoption of guidelines such as those proposed by Geraghty et al.[31], the sex of the original source will have been captured and the use of optimal procedures and a frozen source stock will minimise these challenges.

The SAGER guidelines can also help in more complex experiments to determine whether we need a sex inclusive design. For example, those that have both cell lines and animals within the same study. Consider, indirect in vivo clustered regularly interspaced short palindromic repeat (CRISPR) screens[32] as an example where both primary cells and animals are used within a single study. In these, a two-step process is used where first, in vitro, the target cell line is modified using CRISPR technology and then the modified cell lines are transplanted into an animal. From the perspective of inclusion, this raises the question: should the attention focus on the cell lines or the animals or both? Utilising the principles of the SAGER guidelines, the subject of the research in this study can be identified as the cell line. However, it is known that implanting male cells into female animals has an impact on viability[33]. To be a representative model of human disease, the modified cell lines should be implanted into a wild-type homogenous mouse of the same sex as the cell line. It is important to note that a sex inclusive design is not achieved by just sourcing a second cell line of a similar profile of the other sex. Why? If differences were found between the two cell lines, you would be unable to assign causality as any observed differences could be due to sex-related variation or variation unique to those individual cell lines. To embrace sex inclusion in the design, three or more independently sourced cell lines for each sex would be needed to provide the replication necessary to study the variation arising from sex. Sourcing such cell lines might be prohibitive[29,30].

It is important to clarify, that in addition to replication, inclusive studies require the sexes to be studied at the same time within the same experiment. Sometimes, researchers state they will study another sex later. This is unfortunately, a flawed strategy. When studies are conducted independently, researchers will be unable to assess whether variation in treatment effect is due to sampling variation or associated with the difference in sex.

Many funding bodies[6,34] have released inclusion mandates setting a position that inclusion is the default, and justification is needed for single sex studies. This is providing an impetus for change, but now requires funding reviewers and institute ethical review boards to consider the justifications. It is important to acknowledge that whilst scientists believe sex matters[18], for over 30 years, researchers have been highlighting the embedded bias towards single sex studies in early research[9,35,36]. Many of the justifications given for single sex studies are now recognised as culturally embedded misconceptions. For example, female mice are more variable and therefore their inclusion would increase the number of animals needed[37,38], sex differences in the baseline or sex-related variation in the treatment effect will introduce variability and this would decrease sensitivity to detect the treatment effect of interest[38,39] or inclusion would require a doubling of the sample size[18,40,41]. Many of these misconceptions arise from trying to use the

original statistical toolkit and failing to appreciate that the design has changed, which leads to a concurrent change in the analysis strategy. By embracing the right statistical toolkit, the perceived challenges can be addressed (see Statistical analysis section). Frequently, the resistance to inclusion is that historic research has only been collected on one sex and researchers argue that they wish to compare the results from current to historic. Inclusive research does not hinder a cross comparison of new data to historic in terms of the behaviour; for example, the direction of the effect. The resistance probably represents fear of change and the unknown, but to ensure the science is generalisable, it is necessary to step into the unknown. Inherently, with the single sex workflow is an assumption that the results will generalise. Now, inclusive research will assess the truth of this assumption.

A recent publication has shared the Sex Inclusive Research Framework (SIRF)[42]; a toolkit for both evaluators and scientists to evaluate research proposals. Through a decision tree of twelve questions, proposals are assessed, and a traffic light outcome indicates whether a proposal is appropriate, risky, or insufficient with regard to sex inclusion. For each question, there is supporting information for reviewers conducting the evaluation and educational information to guide the researcher. The idea behind the toolkit is to encourage researchers to reflect on their thinking and to move the discourse to a situation where researchers truly reflect on whether an inclusive design is possible.

**Being clear about the objective of inclusion**. For robust ethical research, it is necessary to align the design, analysis and objectives before conducting the experiments. Studies have found that researchers believe that sex matters as a biological variable[9,41,43] and would like to include males and females as this would allow an exploration of whether there was sex-related variation in the treatment effect however, they do not feel inclusion is doable[18,40]. It is likely that lack of clarity over the objective of inclusion will also acts as a barrier as the objective will impact the design that needs to be implemented, and the resulting number of test samples needed.

Much in vivo research will be exploring the effect of a treatment (the intervention applied) and the goal of including females and males is to assess for the generalisability of the treatment effect. In these situations, the experiment would only be powered to see very large differences in the sex-related variation in the treatment effect[16]. From the perspective of sex inclusion, this should be described as an 'exploratory approach' as sex is not a primary variable of interest. For these studies, the number of animals needed for the effect size of interest can be shared across the males and females provided the right statistical methods are utilised to correctly account for variation in the data[16].

Alternatively, sex could be included in addition to the treatment of interest because sex differences in the outcome is the primary focus. These situations will arise when there is an evidence-based rationale for hypothesized sex difference and therefore should be considered as a 'confirmatory approach'. In these situations, a bespoke power calculation will be needed, considering the differences in the size of the effect. This is an area where more guidance and resources are needed to support the community in appropriately implementing these designs.

Once the sex differences are confirmed and assessed as biologically significant, opportunities arise to explore the underlying biological origins of the observed sex-related differences. Becker et al.[44] have developed a logical series of experimental questions that can be used to guide the research activities to explore the origins of sex-related variation in the treatment effect. For example, step two focuses on assessing if the difference can be attributed to sex hormones. In addition, the manuscript provides advice on potential methods that can be used initially in laboratory studies and then in human studies.

**Should the N be balanced between the sexes?**. A common question when thinking about sex inclusive designs is whether to use a balanced design or a design that represents the disease distribution characteristics. For most situations, a balanced design is appropriate because it ensures a conclusion space which has equal confidence for the sexes tested. An exception could arise when studying a disease which has a very large difference in the sex-specific disease prevalence such that it is almost absent in one sex. For example, if the disease only occurs in 0.1% of the population of men, in these situations, then it becomes a cost-benefit reflection of whether the benefit of inclusion of males is worth the costs (logistic, financial etc.). Drobniak et al. have also identified that an unbalanced design might be necessary if the variability in the outcome measure is very dependent on the sex[45].

**Statistical analysis**. As researchers embrace inclusive designs, whether confirmatory or exploratory, the analysis strategy needs updating to reflect the more complex designs. Unfortunately, errors in the analysis and subsequent interpretation of the results are common (Table 1)[11,46]. Research has shown that incorrect analysis leads towards a positive bias of finding sex differences[11]. Maney and Rich-Edwards, have highlighted how guidance provided by many funding agencies and publishers is encouraging inappropriate analysis and does not align with sound statistical practice[47]. Similarly, an evaluation of three publicly available online training courses for sex inclusive research found they were endorsing invalid analytical approaches that produce false calls[48]. These findings demonstrates the scale of the misunderstandings within the community. The subsequent published research is therefore at high risk of having low reproducibility, which will delay scientific advancement and could undermine the progress being made towards inclusive research that adds value for people of all sexes. There is, therefore, a critical need to engage with applied statisticians in the development of further resources and making community-wide training available to enable researchers to appropriately analyse their data[49].

The exact analysis needs will depend on the research goals, the design and the data characteristics. What is important is that there is a suitable analysis plan that implements a statistically appropriate strategy, which also includes a statistical test that assesses whether there is significant sex-related variation in the treatment effect and accounts for variation arising from sex. The latter increases sensitivity by accounting for a baseline sex difference. The former accounts for variation which increases sensitivity but also statistically assesses whether the treatment effect depends on sex which is an important element of the sex inclusive research vision. This is the recommended strategy, even when you are not powered for detecting an interaction effect, because the statistical power will pass from the main effect to the interaction term when there is a large difference in the effect[16]. For many situations, you will need to embrace a factorial analysis where sex is a factor that could potentially interact with other factors in the dataset (e.g., treatment, age, strain). Figure 1 provides a layman's description of factorial analysis and explains why embracing these more sophisticated analysis tools will maintain sensitivity and why the sample size can be shared between the males and females. Whilst these methods are more statistically sophisticated, most statistical packages will conduct the analysis for the researcher. All that is required is to set up the analysis appropriately. What is more advanced, is interpreting the output with a two-step focus. First, the researcher should inspect the model output for a significant interaction, and then, if significant, focus attention on how sex explained variation in the treatment effect.

### Table 1 | Common errors in the statistical analysis of sex inclusive preclinical studies

| Error | Why? |
|---|---|
| No statistical tests used to support conclusions reached[11]. | Humans are hard-wired to see patterns and therefore statistical tests are needed to challenge cognitive biases and assess whether the observed relationship is caused by something other than chance. |
| Pooling the data for a treatment across the sexes studied[11,16] | Fails to account for variation introduced by sex which will reduce sensitivity and does not allow an assessment of whether the treatment effect depends on sex. |
| Disaggregating the data by sex during the analysis[11,16] e.g., running independent statistical analysis for each sex studied | Loses statistical power as data is not shared in a common statistical model. Does not allow an assessment of whether the treatment effect depends on sex and encourages the comparison of $p$ values error (Differences in sex-specific significance error). |
| Differences in sex-specific significance error (DISS error)[11,46] e.g., finding a statistically significant effect in one sex but not in the other sex and concluding the effect depended on sex. | This is flawed reasoning and is not utilising a statistical test to support the conclusion. This strategy is at high risk of false positives. The differences between significant and not significant could be a function of sampling variability or statistical power. Fundamentally, "because the difference between significant and not significant need not itself be statistically significant"[61]. |
| Comparing the females and males within the treatment group to assess whether there was sex-related variation in the treatment effect[11]. | This is a flawed strategy. It assumes the differences between the groups is due solely to the treatment effect and could be confounded by a common baseline sex difference. |

Rich-Edwards and Maney[7] elaborated upon a 4Cs framework[34] based on four key steps: Consider, Collect, Characterise and Communicate to guide best practice in sex inclusive research. In the expanded framework, the advice is stratified on whether the interest is in the exploratory or confirmatory inclusion of females and males. Where this framework is particularly strong, is in giving advice and recommendations on the analysis and subsequent presentation of the results.

**Interpretation of the results**. In addition to determining the statistical significance of the main effects and interaction, it is important to consider their biological significance. The assessment will be context-dependent and will need to consider the biology and research goals. The movement towards personalised medicine includes the call for sex-specific treatment for medical conditions[50] and provides a motivating scenario for considering sex differences. Researchers will need to be mindful that findings from animal models may not always translate directly to human patients and biological relevance does not guarantee clinical relevance. A decision to explore the personalisation of clinical treatment by sex will require the effect to be quite different for the female and male samples.

In discussions on the reproducibility crisis, much attention has focused on the poor statistical practice of just using $p$ values as the decider of whether something is interesting or not[51].

Including sex makes the interpretation more complex, as both statistical and biological significance must be considered, along with reflecting on what it means for the biology of interest. Consider a research study where there is a statistically significant sex-related variation in the treatment effect such that the size of the effect is more pronounced in the females compared to the males (Fig. 2). Whether this matters is very dependent on the biology. Based on the researcher's expertise and knowledge of the biological system, they could choose to move forward in the research pipeline by working with an average-sized effect across the males and females or could estimate the effect for each sex individually.

Reflecting on the observation that 60% of continuous variables have a baseline sex difference in a trait[2], it is unsurprising that the effect size can be a different size dependent on the sex studied. This scenario is frequently seen when working with mouse models of disease where following induction of the disease state, the outcome variable which marks the disease, is often at a different point on the scale for the males and females. Depending on the biology, this might represent a more extreme disease or just a baseline

difference for the expression of the disease. When reflecting on the treatment effect, it is important to be more nuanced and reflect both on the biological meaning of the estimated effect sizes and the resulting status of the animals. As highlighted in Fig. 3, focusing on $p$ values alone could lead us awry when considering the impact of a treatment. Intuitively, a significant interaction effect between treatment and sex might suggest that there is a biologically important differences, but as illustrated in Fig. 3, panel B, it could equally be the case that differences in baseline means the treatment outcome is the same in the females and the males. Conversely, an intuitive interpretation of a non-significant interaction might lead one to conclude that the treatment is equally effective for males and females, while Fig. 3 panel C illustrates that the similar effect *size* results in a very different biological outcome.

**Visualisation**. As researchers embrace this new way of working, the community will need to learn how to interpret more complex visualisation. Frequently, scientists utilise bar plots to summarise a dataset[52], which presents the mean signal and typically omits any representation of variability. The popularity of such a graph is driven by the desire for a visualisation that rapidly explains the conclusion narrative of the investigation. Such a desire is the probable driver behind data being pooled. Interactions are statistically assessed by comparing the slopes of change and therefore, 2-way ANOVA graphs typically have a mean line between treatment groups stratified by the second factor. If the lines are not parallel, this suggest (subject to the variance) that the treatment effect depends on the second factor. Fundamentally, it is necessary to be able to visualise the change but also the variation in the data to allow a benchmarking of this change. Garofalo et al., in an article exploring best practice for analysis of interactions, recommend reporting the model estimated marginal means with 95% confidence intervals[53].

To make science feasible, a complex biological space is simplified into a testing space and then decisions about the underlying biology are incrementally made through an experimental process of applying an intervention to standardised groups and observing what happens. This approach has integrated with statistical hypothesis testing resulting in the $p$ value being treated as a gate keeper of truth, returning significant or not significant with infrequent consideration of the size of the effect. The data is then often presented as bar plots[54], implying an absolute truth of effect. In the context of the reproducibility crisis, numerous publications have been highlighting

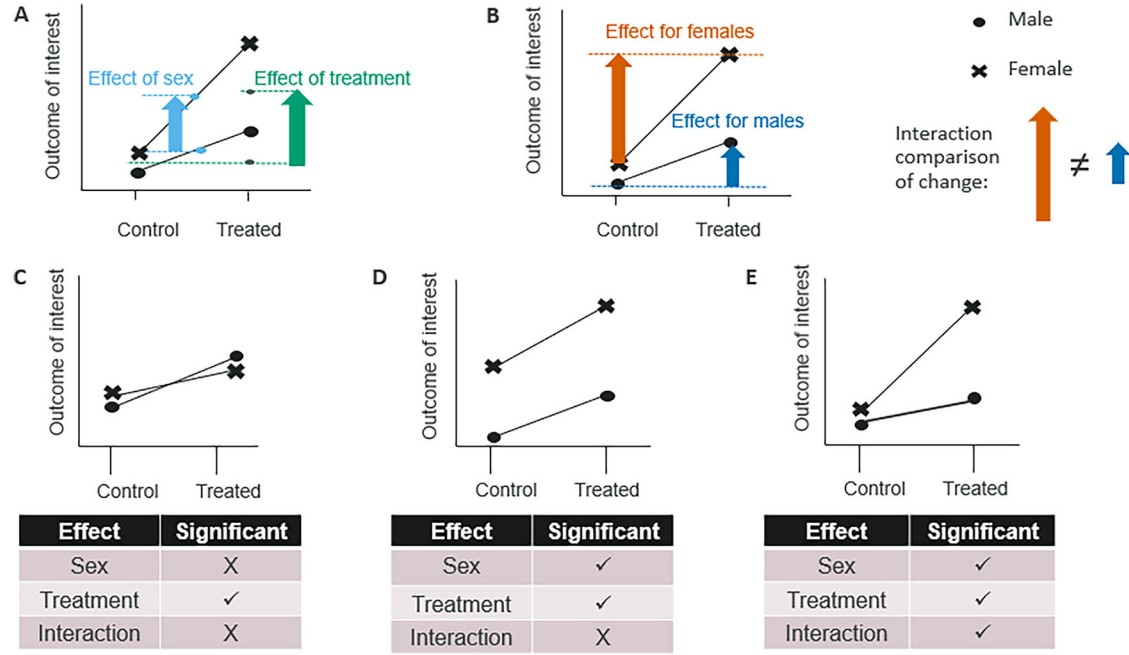

**Fig. 1 | Understanding factorial analysis.** In factorial analysis, a single statistical model is fitted to the data to understand how the different factors influence an outcome metric of interest. The most well-known example, suitable for a continuous normally distributed variable, is a 2-way ANOVA. In such an analysis, the data will be queried with three hypothesis questions testing two main effects (the effect of sex and treatment) and an interaction between sex and treatment. This strategy enhances sensitivity (as the data is shared between the sexes), allows the sex-related variation to be accounted for, and allows a statistical test of whether sex explains variation in the treatment effect. **A** and **B** illustrate how the same data are assessed for main (**A**) and interaction (**B**) effects. **A** The treatment effect is assessed by pooling data across males and females to estimate the average effect of treatment. This is demonstrated by the bluish green arrow which indicates the difference between the control group average (including females and males) and the treatment group average (including males and females). While the effect of sex is mathematically assessed by pooling data across the treatment conditions to estimate the average difference between female and male samples. This is demonstrated by the sky blue arrow, which indicated the difference between the male average (including both control and treated) and the female average (including both control and treated). **B** In the assessment of the interaction, the size of the treatment effect is compared between the females and males. Here, the vermillion arrow indicates the effect of treatment in males, and the blue arrow indicates the effect size of treatment in females. **C-E** Illustrate some of the possible different outcomes from a factorial analysis. **C** An example of a generalizable treatment effect between the females and males where there is no baseline difference between the sexes. **D** An example of a generalizable treatment effect between the males and females in the context of a baseline difference. **E** An example where sex-explains variation in the treatment effect. For both (**C**) and (**D**), the interaction is not significant, and the treatment effect can be assessed solely by looking at the significance of the treatment term in the model and then assessing the estimated treatment effect. For (**E**), the interaction is significant; in these situations, you would run additional analysis to estimate the treatment effect within each sex to understand where the significant differences lie.

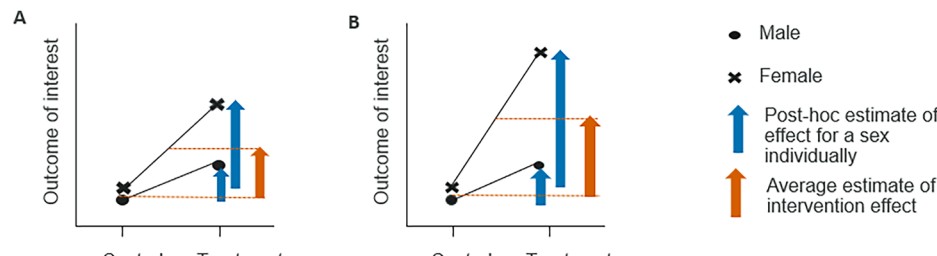

**Fig. 2 | Considering the size of the effect when a treatment effect interacts with sex is critical.** Just because the *p* value is significant for the interaction doesn't mean it is biologically relevant. From the statistical modelling, an average treatment effect across males and females can be estimated (orange arrow) or the treatment effect for each sex can be estimated individually (blue arrows). These effects along with their confidence intervals should be considered. **A** The treatment effect was approximately double the size in the females compared to the males. For this variable this might be considered equivalent and the average effect used. **B** In contrast the treatment effect here is approximately four-fold smaller in the females and in this case the researchers, depending on the biology, might make the call that the individual estimates should be reported, and attention should focus on why the treatment effect was different.

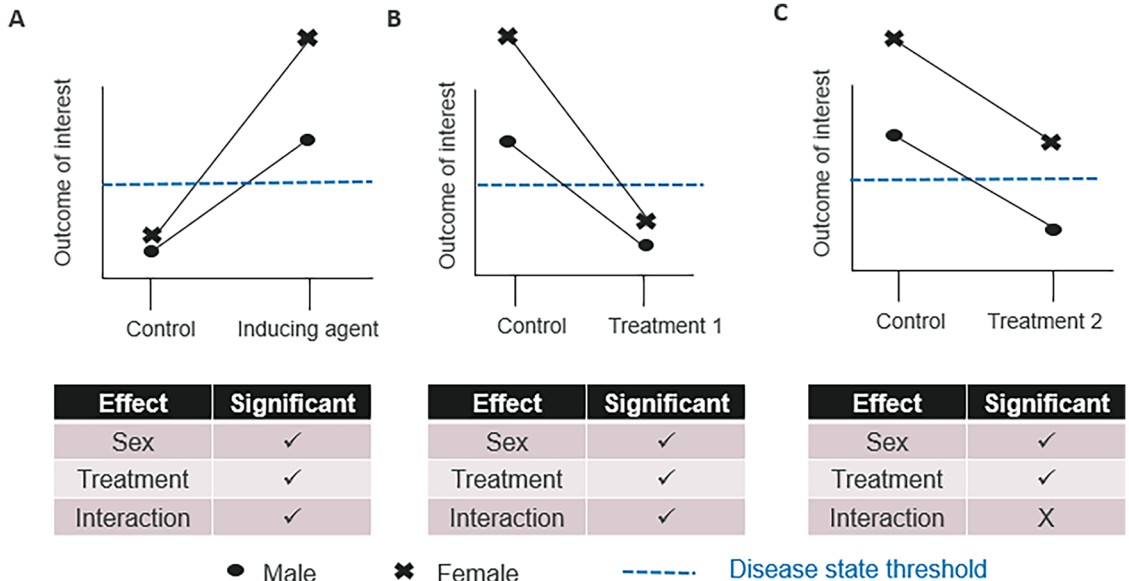

**Fig. 3 | Interpretation requires nuanced reflection considering both statistical and biological significance.** In the following scenario, the need for nuanced interpretation that focuses on the biology of the effect is demonstrated. Graphs plot the marginal means (the average value of a particular group when all other variables are held constant) and the 95% confidence intervals. **A** Consider a model induction scenario, where the inducing agent leads to a change in a critical parameter such that both males and females are in the diseased state (above the blue dotted line) however, the effect size of that effect differs between the two sexes studied. **B** The resulting animals are then treated with treatment 1, which had a statistically significant interaction, such that sex explained variation in the treatment effect. However, in this scenario whilst the effect size is different, the biological impact is the same – both sexes studied are no longer in the disease state. **C** Whilst when the model induced animals are treated with treatment 2, the statistical analysis finds that no significant interaction (the lines are parallel) and estimates a main effect of the treatment. However, in this scenario, whilst the effect size is equivalent the resulting biological state is that the females are still in the disease state whilst the males are not. These hypothetical scenarios demonstrate how important it is to consider the biology in addition to the statistical significance.

that there isn't a pure treatment effect[13,55]. Sex inclusive designs are one element of improving the generalisability of research, but this will require researchers to become more comfortable with biological variation and that there is variation in the treatment response.

It is important to note, that as designs become more complex, researchers will become more reliant on more complex statistical models and then interpretation relies on the estimated effects and their confidence intervals. In these situations, it is important to run the diagnostics to ensure the models represent the data well and to ensure the model is appropriate for the design. Then, the statistical model output and graphs showing the estimated effects with confidence intervals should be shared[53]. As we continue in this journey of inclusive designs, there is a need for further discussion and resources developed to provide advice on best practice for visualising inclusive designs, reporting of effect sizes, and finally, in the data sharing.

**Computational modelling.** Downstream of experiments, data and computer power are used to simulate and study complex systems. The changes around inclusion will therefore have far-reaching consequences. It is, however, early days, but the following examples highlight some opportunities that arise as researchers embrace this new paradigm. Burrowes et al., in a review article considering whole-body physiological in silico models for human health applications, concluded that the consideration of sex differences had been minimal and only started to appear in recent years[56]. They highlight that incorporation of sex requires an understanding of the sex-related differences in biology, physiology and pathophysiology. They argue that inclusion is a critical next step and provide a checklist for appropriate consideration and

inclusion in model development. Furthermore, they highlight that such models will not only incorporate these differences but have the potential to assess the impact on emergent function to further the understanding of differences.

Ontologies allow researchers to categorize observations and identify the key relationships among those concepts. They are an important tool to allow research data to be ingested in a computer-readable format, which allows more complex modelling and exploration of the data across studies to occur[57]. For example, this has been used in complex modelling to conduct comparison across species using data from the International Mouse Phenotyping Consortium by aligning specific disease and or patient phenotypes using the Human Phenotype Ontology and the individual mouse phenotype ontology (Mammalian Phenotype Ontology) which has been used to discover novel disease models for orphan disease[58]. Scientific ontologies are not static as the terms and relationships must evolve to align with the current understanding of the discipline as they adapt to ways in which the researchers are thinking about the research[57]. Currently, the Mammalian Phenotype Ontology is used to classify that the effect is significant in males and females, males alone or females alone. These ontologies will need to evolve to establish methods that can capture thecd relationship between sex and the effect of interest.

Artificial Intelligence (AI) is a key tool with rapid advancement currently occurring that is being used to accelerate research across many disciplines. For example, it is becoming apparent that large language models, a type of generative AI, are introducing game-changing possibilities. As they mine the literature and databases, the quality of the analysis and communication of the results is going to be critical. A common proverb applied to such systems is "you get out what you put in". There is an urgent need to

improve the quality of inclusive research output to ensure the resources using this as input produce output that are valuable.

## Moving beyond the binary construct

In the future, researchers will need to go beyond the binary construct when thinking about sex inclusive research. McLaughlin et al. highlight that in the context of ecology and evolution research with a number of case studies, that an expanded framework to represent sex, which uses multivariate and non-binary variables will better allow an exploration of the biology[26]. Likewise, Sanchis-Segura and Wilcox[59], challenged the use of sex with a binary classification, highlighting the risk in that strategy as a mean comparison. They argue that this may impede advances toward precision medicine and advocate for the selection of continuous variables that represent sex-related variation. Whilst Pape et al. argue that to truly understand sex differences, it is necessary to reflect and select concrete, measurable, sex-related variables which for the system being studied, would provide plausible mechanisms to understand what is driving the sex-related differences[21].

Embracing these strategies will start to unpack the biology and improve understanding and hopefully provide more clinically relevant insights. However, it is important to place this in the context of the research pipeline. As sex differences start to be understood, this paves the way for more nuanced approaches, but at this point in time, for most researchers, the goal of inclusion is to improve generalisability. Arnold et al. raised a concern that critiques of research on sex differences from different viewpoints and a concern over the potential misinterpretation or misuse of findings could undermine the progress on inclusion[60]. This can be avoided by improving communities understanding of the goal of inclusion, the language used, the quality of the analysis, and presentation of the results.

## Conclusions

Over the last decade, the landscape around sex inclusive research has completely changed from a few scientists reporting on an embedded bias to a mainstream topic. There is now tangible change occurring on the ground: the proportion of published work that includes more than one sex has significantly increased[8,9]. This has been driven by active research, discussion, and exploration of the issues by many individuals across the research landscape. Including females and males in preclinical research, impacts the whole research pipeline from planning, analysis, visualisation, communication, modelling etc. and there is much to be done for the benefit to be fully realised. As we navigate these paradigm shifts, new lessons have been learnt and will continue to arise. Consequently, there will be a need to revisit and integrate these learnings into earlier frameworks, guidelines etc. to keep them relevant and up to date.

I would argue, then that sex inclusive research is a success story. We, the research community, are making significant strides forward as we embrace the paradigm shift. As we embrace this new way of working, the rewards will be more representative, replicable conclusions and the identification of opportunities where exploration of variation associated with sex will also illuminate new paths to investigate the underlying biological cause of sex-related variation.

**Reporting summary**. Further information on research design is available in the Nature Portfolio Reporting Summary linked to this article.

## Abbreviations

| | |
|---|---|
| AI | Artificial Intelligence |
| CRISPR | clustered regularly interspaced short palindromic repeat |
| DISS | Differences in sex-specific significance error |
| SAGER | Sex and Gender Equity in Research |
| SIRF | Sex Inclusive Research Framework |

**Natasha A. Karp** [ORCID] ✉

Data Sciences & Quantitative Biology, Discovery Sciences, R&D, AstraZeneca, Cambridge, UK. ✉e-mail: Natasha.Karp@astrazeneca.com

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

## Acknowledgements
NAK would like to acknowledge the use of a Large Language Model to provide grammar suggestions during the writing process. Funding source: This project received no external funding.

## Competing interests
The author declares the following competing interests: NAK is an employee of AstraZeneca and has shareholdings in AstraZeneca.
