## [Transparent Peer Review file · Communications Biology]

Navigating the Paradigm Shift of Sex Inclusive Preclinical Research and Lessons Learnt

Corresponding Author: Dr Natasha Karp

Version 0:

Reviewer comments:

Reviewer #1

(Remarks to the Author)

This is an important manuscript that engages with a topic that has received a lot of attention recently. The field of sex differences (and how to treat sex as a variable in research) is at a crossroads, with many voices arguing for more than a decade now that sex bias in research is still a problem, and many newer voices calling for caution. There is thus an important space to be filled here – the melding of these two viewpoints in a way that reaches both audiences and finds common ground.

There is a risk that instead of forging that common ground, such efforts will instead appear as a sort of Frankenstein, combining the two conflicting viewpoints instead of reconciling them. This manuscript could be falling into that space. In some places, it aligns with older NIH/SAGER arguments about why sex is the most important and only variable that all research must consider. The emphatic distinction between sex and gender (sex = "biological") also seems to align with more traditional thinking. In other sections, the manuscript endorses more forward-thinking viewpoints, for example discussing the non-binary nature of sex, importance of inclusive language, etc. These perspectives are at odds with each other. I think this manuscript could be quite impactful if it could move a bit further toward reconciling them, ideally by being more explicit about where the older framework needs to be updated.

An example appears in Row 2 of Table 1, where it is argued that pooling data fails to consider variability explained by sex category. I think there is a real need to consider (and provide a convincing argument about) why sex category is the only variable being singled out as the most important factor that populations and samples must be divided into. Pooling fails to account for all variation, not just variation explained by sex, and there are countless other variables that explain variation better than sex category. Why single out sex? Rather than trying to justify the choice of sex as the only important variable, it might be good (and honest) to simply say that the reason sex is being singled out is that NIH says so, and to critique that choice (as the manuscript does, in other places).

Parts of the manuscript rely heavily on SAGER, which is problematic with respect its endorsement of invalid statistical methodologies, stark conceptual separation of sex and gender, and framing of sex as binary. In these sections, it would be good to take a more critical stance toward SAGER, acknowledging that it is important historically but that it is a flawed framework.

Other points:

L 72 I'm not sure using sex "as a verb" is what is meant here. Perhaps "have sex" is what was intended?

L 72-74, might want to make clear these are definitions used by NIH, etc. and not universally accepted in all fields. I would perhaps avoid "biological" which is rapidly becoming associated with denial of rights to marginalized people.

Table 1, I've never totally understood NIH's definition of "disaggregation" but I think data can be disaggregated and still included in the same model. So here maybe say "separated for analysis" or something like that. I wonder if "stratified" could be mentioned which is also a word with many meanings to different people.

Box 1, I think usage of red for both sex and female and blue for both treatment and male is a little confusing. Perhaps using two different colors in panel A would be better.

Box 1 is introduced as 'more sophisticated analysis' with five panels and five paragraphs of text, which could be discouraging to researchers in a hurry. So it might be a good idea to explain up-front that most statistical packages will do this analysis for the researcher – in most cases it is not necessary to do anything very sophisticated to set up a two-way ANOVA. The trick is interpreting the output, particularly inspecting the result for the interaction before interpreting and presenting post-hoc pairwise comparisons. I think a brief "how to interpret the output of your stats package" might be an excellent addition here and could be done with only a sentence or two, just connecting what is typically spat out (e.g. p value for "main effect," etc.) with the language in panels C-E.

L 286: might want to avoid term "sex effect" if also arguing against thinking of sex category as a causal variable (L96) or very briefly explain why the term is being used here.

L297, I found it confusing to refer to a "previous section" when I believe it is the one immediately above, right? So this opening might be better as "In addition to determining statistical significance of the effects and interactions, it is important to consider their biological significance."

It would be helpful to frame this section by referring to calls for sex-specific treatments for medical conditions. Such practices could only be beneficial if sex differences are quite large. I think it's particularly important to emphasize the real-world consequences of the over-interpretation of sex differences, given that parts of this section are pretty dense (e.g., L328-342 and the caption for Fig. 2) and so need to be framed in terms of impact to motivate readers to engage with what is being argued here.

L302, make clearer that the "second factor" is sex. As written, it seems like the second factor is biological significance (the first being statistical significance).

Reviewer #2

(Remarks to the Author)

The major claims of the paper are important and of broad impact. There have been numerous calls for increased sex-inclusiveness of biomedical research and this paper outlines that relevant history. The paper incorporates how language as well as its imprecise uses, e.g., 'both sexes' reveal limited understanding and binary perspectives embedded in our current practices. How sex is operationalized is also addressed, getting to the root of the value of a sex-inclusive approach. The paper also addresses modern technological advances such as CRISPR and the complexity of working with cell lines where chromosomes may be lost or duplicated. A recent approach, the Sex Inclusive Research Framework toolkit is also featured-an important advance. The issues of study design and analyses are extensively discussed as is the challenge in invalid analytical approaches and limited understanding of appropriate sex-inclusive study designs. The Table of Common Errors is particularly useful. The figures are equally informative and the importance of data visualization is addressed well.

Reference 7 is cited as having 'published a 4 Cs framework' which was in fact developed by the NIH Office of Research on Women's Health and was instead utilized as an organizing principle in their paper. This misattribution should be corrected and the 4 Cs framework appropriately cited.

Version 1:

Reviewer comments:

Reviewer #1

(Remarks to the Author)

The author has adequately addressed all my concerns. I have only one very minor point which is that changing "sex effect" to "effect of sex" doesn't solve the original problem. The issue is that if sex is not causal, then it technically cannot have an "effect". But when talking about statistical models and reporting results, even in pseudo-experiments we still use the term "effect." So my suggestion was just to briefly explain that.

It is a great manuscript and I look forward to seeing it published.

Reviewer #2

(Remarks to the Author)

482 Embracing these strategies will start to unpick the biology and improve understanding and
483 hopefully provide more clinically relevant insights.

Is 'unpick' meant to be used here? Perhaps 'unpack' was intended?

The authors have addressed many of the concerns identified in review and this has improved the manuscript. By confining

the scope to pre-clinical research, the issues raised about gender have largely been rendered minimal. The reliance on biological relevance of differences is one result of this limitation. Biologically relevant findings may not be clinically meaningful for one sex as an entire group, e.g. all males, or may not be clinically meaningful for every person of a give sex, e.g. a particular female human. I wonder if this perspective could be incorporated as this sentence could be interpreted to suggest that biologically relevant equals clinically relevant and that relevance for members of one sex (as in a large treatment effect difference) is necessary for personalized medicine. Moving from statistically significant treatment effects in an animal or preclinical model to the realities of treating individual patients is buried here. I know this manuscript is focused on pre-clinical research but to the large extent that such efforts both expand our understanding of biology, and pathology, and inform translational efforts that may eventually affect clinical care, I think that this nuance is important to include.

"The movement towards personalised

333 medicine includes the call for sex-specific treatment for medical conditions (54) and provides
334 a motivating scenario for considering sex differences. However, a decision to explore the
335 personalisation of clinical treatment by sex will require the effect to be quite different for the
336 female and male samples. I".

I thank the reviewers for their careful review of the material and have addressed the feedback below.

Reviewer #1 (Remarks to the Author):

This is an important manuscript that engages with a topic that has received a lot of attention recently. The field of sex differences (and how to treat sex as a variable in research) is at a crossroads, with many voices arguing for more than a decade now that sex bias in research is still a problem, and many newer voices calling for caution. There is thus an important space to be filled here – the melding of these two viewpoints in a way that reaches both audiences and finds common ground.

I appreciate the reviewer's comment and take heart that they appreciate the article goal of finding common ground and presenting the progress as a success story.

There is a risk that instead of forging that common ground, such efforts will instead appear as a sort of Frankenstein, combining the two conflicting viewpoints instead of reconciling them. This manuscript could be falling into that space. In some places, it aligns with older NIH/SAGER arguments about why sex is the most important and only variable that all research must consider. The emphatic distinction between sex and gender (sex = "biological") also seems to align with more traditional thinking.

The feedback is an interesting perspective. After reviewing the material, I have looked to address this concern in a number of ways.

Firstly, I have endeavoured to elevate that these topics are being discussed in the context of preclinical research. I have added this to the title, abstract and a number of positions in the main manuscript. This is relevant because in the context of preclinical research we cannot explore the human experience of gender and therefore can only discuss sex. The objective in this article is to encourage researchers to understand the limits of their inference space and correctly describe their science. It is not a commentary of the distinction between sex and gender within humans.

Then, I have added to the introduction, a section to highlight that here is debate around sex-inclusion and whether other variables should be prioritised.

In other sections, the manuscript endorses more forward-thinking viewpoints, for example discussing the non-binary nature of sex, importance of inclusive language, etc. These perspectives are at odds with each other. I think this manuscript could be quite impactful if it could move a bit further toward reconciling them, ideally by being more explicit about where the older framework needs to be updated.

I appreciate the reviewer recognising the forward-thinking viewpoint. I have extended the conclusions to provide a reflection that as lessons are learnt there will be a need to revisit and update the advice included in earlier frameworks/guidelines. I have not been explicit on which need revisiting as this paper is arguing that we are in mist of change and hence will be impacting all resources developed.

An example appears in Row 2 of Table 1, where it is argued that pooling data fails to consider variability explained by sex category. I think there is a real need to consider (and

provide a convincing argument about) why sex category is the only variable being singled out as the most important factor that populations and samples must be divided into. Pooling fails to account for all variation, not just variation explained by sex, and there are countless other variables that explain variation better than sex category. Why single out sex? Rather than trying to justify the choice of sex as the only important variable, it might be good (and honest) to simply say that the reason sex is being singled out is that NIH says so, and to critique that choice (as the manuscript does, in other places).

Regarding the inclusion in Table 1, which is identifying common errors in statistical analysis of sex inclusive preclinical studies. I have amended the figure title to be explicit that this is in the context of preclinical studies.

In the preclinical context, the researcher has significant control over the design of the experiment and the sources of variation included in the experiment. Consequently, when discussing inclusive designs, we are talking about moving from highly standardised experiments with only one sex, one genetic background, one age etc to two sexes. Research has shown that many researchers, when they are transitioning into these factorial designs pool the data. Table 1 is focused on these common errors, and it is in the context of a factorial design where the only sources of variation are the intervention of interest and sex in an alignment with the sex-inclusive mandates. In this context, I would argue that the text is appropriate.

However, what I appreciate is that the manuscript doesn't join the debate on why sex is being prioritised and whether other variables that explain biological variation should be explored. This was originally a deliberate decision, because this manuscript is focusing on the position of the lessons learnt on sex inclusive research not focused on justifying whether we should or should not focus on sex. I have added to the Introduction a few sentences to clarify that there is some debate on the topic of inclusion and provide additional references to represent this view.

I have amended the title to include the word preclinical to ensure the scope of the material is clear.

Parts of the manuscript rely heavily on SAGER, which is problematic with respect its endorsement of invalid statistical methodologies, stark conceptual separation of sex and gender, and framing of sex as binary. In these sections, it would be good to take a more critical stance toward SAGER, acknowledging that it is important historically but that it is a flawed framework.

One of the underlying concepts of this manuscript is we are working it all out and different communities have different things to bring to the table to refine our thinking. I am arguing we are within a paradigm shift and hence the landscape is changing rapidly. The SAGER guidelines were developed in 2016, and I am therefore not surprised that the reviewer perceives the SAGER guidelines to be important historically but now a framework with some flaws. I, however, don't think this is a unique issue to the SAGER guidelines. I have amended the conclusions to try and raise this issue but feel uncomfortable taking a critical stance towards SAGER when the issue is not unique to SAGER.

Other points:

L 72 I'm not sure using sex "as a verb" is what is meant here. Perhaps "have sex" is what was intended?

Thanks for highlighting the lack of clarity. The text has been amended to 'when used as a shortened form for "sexual intercourse"'.

L 72-74, might want to make clear these are definitions used by NIH, etc. and not universally accepted in all fields. I would perhaps avoid "biological" which is rapidly becoming associated with denial of rights to marginalized people.

The section has been reworded to be more accurate by being explicit that sex is a categorisation mechanism using a set of attributes and the word biological has been removed. I have amended the text to use a definition for gender from Pape et al (2024). I did not find a paper that was explicit that there was disagreement in the definition of sex/gender in science to support a statement that it is not universally accepted in all fields. I have added the subordinate clause "In science," to provide context.

Table 1, I've never totally understood NIH's definition of "disaggregation" but I think data can be disaggregated and still included in the same model. So here maybe say "separated for analysis" or something like that. I wonder if "stratified" could be mentioned which is also a word with many meanings to different people.

I agree there is much confusion around the NIH's definition of disaggregation. I hope they mean it more in the context of reporting summary statistics than the analysis. The intention in the article is to try and help provided clarity. I reworded the section to address your concern. I haven't used the word stratified as it is a technical word without consensus and isn't mainstream terminology for the preclinical target audience.

Box 1, I think usage of red for both sex and female and blue for both treatment and male is a little confusing. Perhaps using two different colors in panel A would be better.

Thank you for raising the issue. Alternative colours have been selected

Box 1 is introduced as 'more sophisticated analysis' with five panels and five paragraphs of text, which could be discouraging to researchers in a hurry. So it might be a good idea to explain up-front that most statistical packages will do this analysis for the researcher – in most cases it is not necessary to do anything very sophisticated to set up a two-way ANOVA. The trick is interpreting the output, particularly inspecting the result for the interaction before interpreting and presenting post-hoc pairwise comparisons.

I appreciate the suggestion to reassure the reader that implementing such an analysis is achievable. The following text has been added to the main text of the article "*Whilst these methods are more statistically sophisticated, most statistical packages will conduct the analysis for the researcher. All that is required is to set up*

the analysis appropriately. What is more advanced, is interpreting the output with a two-step focus. First, the researcher should inspect the model output for a significant interaction, and then, if significant, focus attention on how sex explained variation in the treatment effect. ”

I think a brief “how to interpret the output of your stats package” might be an excellent addition here and could be done with only a sentence or two, just connecting what is typically spat out (e.g. p value for “main effect,” etc.) with the language in panels C-E.

Additional text has been added to the panel as suggested: “For both C and D, the interaction is not significant, and the treatment effect can be assessed solely by looking at the significance of the treatment term in the model and then assessing the estimate treatment effect. For E, the interaction is significant, in these situations, you would run additional analysis to estimate the treatment effect within each sex to understand where the significant differences lie.”

L 286: might want to avoid term “sex effect” if also arguing against thinking of sex category as a causal variable (L96) or very briefly explain why the term is being used here.

I did contemplate what term to use here when writing the original version. With your prompt, and a fresh review, I have removed the term and rephrased the sentence to better align with message of the manuscript. It now reads: *“While the effect of sex is mathematically assessed by pooling data across the treatment conditions to estimate the average difference between female and male samples”*

L297, I found it confusing to refer to a “previous section” when I believe it is the one immediately above, right? So this opening might be better as “In addition to determining statistical significance of the effects and interactions, it is important to consider their biological significance.”

Amended as suggested

It would be helpful to frame this section by referring to calls for sex-specific treatments for medical conditions. Such practices could only be beneficial if sex differences are quite large. I think it’s particularly important to emphasize the real-world consequences of the over-interpretation of sex differences, given that parts of this section are pretty dense (e.g., L328-342 and the caption for Fig. 2) and so need to be framed in terms of impact to motivate readers to engage with what is being argued here.

Thank you for the suggestions to provide more context to this discussion. This has been added to the introduction paragraph and includes the following *“In addition to determining statistical significance of the effects and interactions, it is important to consider their biological significance. The assessment will be context dependent and will need to consider the biology and research goals. The movement towards personalised medicine includes the call for sex-specific treatment for medical conditions (54) and provides a motivating scenario for considering sex differences.*

However, a decision to explore the personalisation of clinical treatment by sex will require the effect to be quite different for the female and male samples.”

L302, make clearer that the “second factor” is sex. As written, it seems like the second factor is biological significance (the first being statistical significance).

Thank you for highlighting. The phrase second factor has now been replaced with the word sex.

Reviewer #2 (Remarks to the Author):

The major claims of the paper are important and of broad impact. There have been numerous calls for increased sex-inclusiveness of biomedical research and this paper outlines that relevant history. The paper incorporates how language as well as its imprecise uses, e.g., 'both sexes' reveal limited understanding and binary perspectives embedded in our current practices. How sex is operationalized is also addressed, getting to the root of the value of a sex-inclusive approach. The paper also addresses modern technological advances such as CRISPR and the complexity of working with cell lines where chromosomes may be lost or duplicated. A recent approach, the Sex Inclusive Research Framework toolkit is also featured- an important advance. The issues of study design and analyses are extensively discussed as is the challenge in invalid analytical approaches and limited understanding of appropriate sex-inclusive study designs. The Table of Common Errors is particularly useful. The figures are equally informative and the importance of data visualization is addressed well.

I appreciate that the reviewer found this manuscript to be important, useful and that it would have broad impact.

Reference 7 is cited as having 'published a 4 Cs framework' which was in fact developed by the NIH Office of Research on Women's Health and was instead utilized as an organizing principle in their paper. This misattribution should be corrected and the 4 Cs framework appropriately cited.

The additional reference has been added and the text amended to ensure that it clear that Rich-Edwards and Maney built upon the 4Cs framework.

REVIEWERS' COMMENTS:

Reviewer #1:

1. The author has adequately addressed all my concerns.

That is good to hear

2. I have only one very minor point which is that changing "sex effect" to "effect of sex" doesn't solve the original problem. The issue is that if sex is not causal, then it technically cannot have an "effect". But when talking about statistical models and reporting results, even in pseudo-experiments we still use the term "effect." So my suggestion was just to briefly explain that.

Thank you for the push to be more explicit around terminology and clarify the use of the word effect when conducting statistical analysis. I have amended the text in the manuscript to read: "As research practices become more inclusive and identify situations where sex mathematically explains variation in the treatment effect, it is important to be mindful of the trap of perceiving sex as an underlying causal mechanism (21). When conducting statistical analysis, the term effect is used for the various element of the model which implies causality but in the context of statistically modelling means that the variation associated with this term is explored"

3. It is a great manuscript and I look forward to seeing it published.

I appreciate the feedback and refinement advice given

Reviewer #2 (Remarks to the Author):

"Embracing these strategies will start to unpick the biology and improve understanding and hopefully provide more clinically relevant insights." Is 'unpick' meant to be used here? Perhaps 'unpack' was intended?

For me the terms are synonyms, but I understand that unpack is more widely accepted terminology in academic discourse, so I have changed the word to align.

The authors have addressed many of the concerns identified in review and this has improved the manuscript. By confining the scope to pre-clinical research, the issues raised about gender have largely been rendered minimal.

Pleased the clarification has addressed the concerns

The reliance on biological relevance of differences is one result of this limitation. Biologically relevant findings may not be clinically meaningful for one sex as an entire group, e.g. all males, or may not be clinically meaningful for every person of a give sex, e.g. a particular female human. I wonder if this perspective could be incorporated as this sentence could be interpreted to suggest that biologically relevant equals clinically relevant and that relevance for members of one sex (as in a large treatment effect difference) is necessary for personalized medicine. Moving from statistically significant treatment effects in an animal or preclinical model to the realities of treating individual patients is buried here. I know this manuscript is focused on pre-clinical research but to the large extent that such efforts both expand our understanding of biology, and pathology, and inform translational efforts that may eventually affect clinical care, I think that this nuance is important to include.

"The movement towards personalised medicine includes the call for sex-specific treatment for medical conditions (54) and provides a motivating scenario for considering sex

differences. However, a decision to explore the personalisation of clinical treatment by sex will require the effect to be quite different for the female and male samples. "

The following sentence has been added to provide attention context to temper the thinking. "Researchers will need to be mindful that findings from animal models may not always translate directly to human patients and biological relevance does not guarantee clinical relevance."